# Extraction of Polyphenolic Antioxidants from Red Grape Pomace and Olive Leaves: Process Optimization Using a Tailor-Made Tertiary Deep Eutectic Solvent

**Vassilis Athanasiadis \*** , **Dimitrios Palaiogiannis, Konstantina Poulianiti** , **Eleni Bozinou** , **Stavros I. Lalas** and **Dimitris P. Makris**

Department of Food Science & Nutrition, University of Thessaly, N. Temponera Street, 43100 Karditsa, Greece; dipaleog@med.uth.gr (D.P.); kpoulianiti@uth.gr (K.P.); empozinou@uth.gr (E.B.); slalas@uth.gr (S.I.L.); dimitrismakris@uth.gr (D.P.M.)
\* Correspondence: vaathanasiadis@uth.gr

**Abstract:** In the framework of introducing green strategies for food processing, the industrial orientation has shifted towards the replacement of conventional petroleum-based solvents with alternative eco-friendly ones. On this basis, the objective of this study was to synthesize a novel, tertiary, food-grade deep eutectic solvent, composed of glycerol, citric acid, and L-proline (GL-CA-Pro), and to test it as a solvent for the extraction of polyphenols from agri-food waste biomass. After an initial screening on various common residual materials (apple peels, lemon peels, orange peels, red grape pomace, olive leaves), evidence emerged that indicated GL-CA-Pro was more effective than other DESs commonly used for polyphenol extraction. Furthermore, extracts from red grape pomace (RGP) and olive leaves (OLL) were shown to contain higher level of total polyphenols and increased antioxidant activity. Process optimization for those two materials with the response surface methodology revealed that the major difference pertained to the extraction time. In addition, for both materials, GL-CA-Pro was shown to provide higher total polyphenol yields (53.25 and 42.48 mg gallic acid equivalents per g of dry mass, respectively) compared to water and 60% aqueous ethanol. However, the chromatographic analyses for OLL suggested aqueous ethanol was a more suitable solvent for some principal polyphenolic constituents. The RGP extract produced with GL-CA-Pro exhibited significantly stronger antioxidant effects compared to the aqueous and hydroethanolic extracts, but the outcome for the OLL extracts was diversified. It was concluded that GL-CA-Pro is a very efficient solvent for RGP polyphenols, but its efficiency regarding OLL was comparable to that of aqueous ethanol.

**Keywords:** antioxidants; deep eutectic solvents; extraction; olive leaf; polyphenols; red grape pomace

## 1. Introduction

Worldwide, the agricultural sector is a source of a high volume of by-products and wastes, which are derived from cultivation and farming practices, as well as post-harvest losses and processing. It has been estimated that almost 140 billion tons of agri-food biomass is generated globally on an annual basis, embracing materials such as wood, bark, roots, leaves, stalks, bagasse, straw residues, seeds, and peels [1]. Agri-food wastes cause disposal problems with potentially harmful consequences to the environment, and such residues have been traditionally used mainly as fertilizers and animal feed. However, the increasing awareness for environmental pollution risks, as well as the need for the conservation of resources, has triggered the development of technologies for more efficient agri-food waste valorization, and the production of value-added chemicals, food/pharmaceutical/cosmetic ingredients, and biofuels [2].

Specifically for the Mediterranean basin, it has been outlined that policies for agri-food waste valorization should mainly focus on the wine and olive oil industries. In both cases,

the processing residues generated may contain substances of interest to the food, pharmaceutical, and cosmetics manufacturing industries [3]. The winemaking process yields large amounts of vine shoots and vinification by-products, such as grape pomace and stalks. Production of olive oil is associated with by-products such as olive leaves and liquid wastes, which are mainly represented by the olive mill wastewater (OMW). All these side streams are materials that are particularly rich in polyphenolic phytochemicals, encompassing simple phenolics (e.g., tyrosol, gallic acid), polyphenolic derivatives (e.g., oleuropein), and a large spectrum of flavonoids (flavones, flavanols, flavonols, anthocyanins). Thus, by virtue of their high volume, abundance, and richness in polyphenolic compounds, winemaking and olive oil production residues are exceptional sources of high value-added materials [2].

Polyphenols are natural secondary plant metabolites that have been found to express numerous chemical and/or biochemical mechanisms, through which these molecules may manifest their effects [4]. Polyphenols may act as antioxidants, reactive oxygen species (ROS) scavengers, and metal ion reducers or chelators. Moreover, inhibitory effects of polyphenols towards enzymes related to disorders, such as type 2 diabetes and obesity (e.g., $\alpha$-amylase, $\alpha$-glucosidase, etc.), have also been well documented. Furthermore, polyphenols may exhibit anti-inflammatory and antimicrobial effects [4]. Therefore, it is widely accepted that polyphenolic substances may be the most important non-nutrient bioactive compounds in the human diet.

Polyphenols can be very efficiently recovered from various plant materials by deploying solid–liquid extraction processes, which are traditionally performed with solvents of appropriate polarities (e.g., water/ethanol or water/methanol mixtures). Along with the development of novel, high-performance, green extraction technologies (i.e., ultrasound-assisted, pulsed electric field, etc.), a great effort has also been expended on the search for alternative, green, and effective solvents for polyphenol extraction [5,6]. Mainly over the last seven years, there has been a boost in the literature concerning the use of innovative liquids, termed as deep eutectic solvents (DESs), in processes developed for polyphenol extraction. DESs usually comprise two constituents, one functioning as a hydrogen bond donor and the other as a hydrogen bond acceptor [7,8].

These solvents can easily be synthesized by naturally occurring constituents, such as polyols, amino acids, sugars, organic acids, and salts thereof. In several cases some DESs have been proven to significantly outperform commonly used, conventional solvents [9]. The advantages that DESs exhibit over petroleum-based solvents is their recyclability, very low vapor pressure, absence of toxicity, compatibility with foods/pharmaceuticals/cosmetics, and their exceptional solvation properties. These features make DESs unique solvents for the implementation of eco-friendly extraction strategies.

On this ground, the main objectives of this study were as follows: (i) to synthesize a novel, tailor-made, tertiary DES, based on a rational design; (ii) to screen various agri-food wastes in order to obtain some basic information regarding the efficiency of the novel DES to extract polyphenolic compounds; and (iii) to optimize the extraction process for the effective recovery of polyphenolic compounds for the richest materials.

## 2. Materials and Methods

### 2.1. Chemicals and Reagents

2,2-Diphenyl-1-picrylhydrazyl (DPPH) was purchased from Alfa Aesar (Karlsruhe, Germany). L-Ascorbic acid and citric acid were obtained from Carlo Erba (Milano, Italy). Glycerol anhydrous and sodium carbonate anhydrous were from Penta (Prague, Czech Republic). 2,4,6-Tris(2-pyridyl)-s-triazine (TPTZ) was from Fluka (Steinheim, Germany). Iron (III) chloride hexahydrate (FeCl$_3$) was from Merck (Darmstadt, Germany). Absolute ethanol, methanol, Folin-Ciocalteu regent, gallic acid monohydrate, and hydrogen chloride (37%) were from Panreac (Barcelona, Spain). Catechin, quercetin, caffeic acid, luteolin 7-*O*-glucoside, hydroxytyrosol ($\geq$98%), oleuropein (98%), and apigenin were from Sigma-Aldrich (Steinheim, Germany). L-Proline was from Glentham Life Sciences (Corsham, UK). Pelargonin (pelargonidin 3,5-di-*O*-glucoside) chloride was obtained from Extrasynthese

(Genay, France). The deionized water used in the experiments was produced using a deionizing column. All solvents used for chromatography were HPLC grade.

## 2.2. Synthesis of DES

The DES utilized in this study was made according to a published technique [10]. Briefly, glycerol (a hydrogen bond donor, HBD) was mixed with citric acid and L-proline (a hydrogen bond acceptor, HBA) at a molar ratio of 2:1:1. The mixture was put into a round-bottom glass flask and heated gently (75–80 °C) for 120 min until a perfectly transparent liquid was formed. An oil bath was used to heat the room, which was placed on a hotplate with a thermostat (Heidolph MR Hei-Standard, Heidolph Instruments GmbH & Co. KG, Schwabach, Germany). The liquid, termed as GL-CA-Pro, was allowed to reach room temperature before being stored in the dark in a sealed vial. Over the course of six weeks, the appearance of crystals that could suggest instability was examined at regular intervals. The DESs, which were composed of glycerol and choline chloride at a HBD/HBA molar ratio of 2 and glycerol and tri-sodium citrate at a HBD/HBA molar ratio of 15, were synthesized in the same way and termed as GL-ChCl and GL-TSC, respectively.

## 2.3. Waste Materials and Handling

The olive leaves (OLL) were from the Agrielia Kalamon variety (*Olea europaea* L.) and were harvested from an olive tree plantation in the region of Avlida (Evia, Central Greece), using a method that ensured minimal composition change due to differences in sunshine exposure [11]. The leaves were brought to the laboratory within 3–4 h of harvesting and then dried for 24 h at 55 °C in a laboratory oven (Binder BD56, Bohemia, NY, USA). Then, using a blender (Camry, Poland), they were pulverized to particle sizes (using a Fritsch Analysette 3, Idar-Oberstein, Germany) of around 207 μm and kept in the freezer (−40 °C).

The red grape pomace (RGP) was made using grapes (*Vitis vinifera* cv. Muscat of Hamburg) that were kindly donated by a winery in Karditsa (Central Greece). After a 7-day pomace interaction, the waste was collected, transported to the laboratory within 2 h, and stored at −40 °C. A quantity of RGP was thawed and distributed on metal trays in layers of roughly 0.5 cm thickness to prepare the required material for solid–liquid extraction. The trays were dried for 6 h at 80 °C in a laboratory oven. The powder was produced by pulverizing the dry material (2% moisture content) into a blender, sieving it to produce a powder with an average particle diameter of around 110 μm, and then storing it at −40 °C.

The rest of the agri-food wastes tested (lemon peels, orange peels, apple peels) were obtained from local catering facilities, a few hours after the processing of the corresponding fruit. The conditions regarding handling and processing were identical to those described for RGP.

## 2.4. Extraction Procedures

All extractions were carried out using 10 mL of solvent and 1 g of dry mass (DM) (with a constant liquid-to-solid ratio of 10 mL g$^{-1}$). The solvent and the dried material were transferred into a 25-mL Duran™ glass bottle immersed in an oil bath, and magnetic stirring, as well as heating at the desired temperature, were accomplished by a temperature-controlled hotplate (Witeg, Wertheim, Germany), operated at 500 rpm. For the preliminary extractions of various agri-food wastes, the extractions were accomplished at 50 °C. For the response surface methodology, the resident time ($t$) and extraction temperature ($T$) were adjusted in accordance with the experimental design. The initial screening was carried out on various plant by-products using all DESs at a proportion of 8/2 (*w/w*) with water, at a constant temperature of 50 °C, for 180 min. Control extracts with deionized water and 60% (*v/v*) ethanol were prepared at 70 °C, and resident times of 60 and 180 min, respectively. These conditions represented average values of extraction conditions reported in a previous extensive study [12].

### 2.5. Design of Experiment–Response Surface Methodology Optimization

The objective of the experimental design was the maximization of the extraction yield in total polyphenols ($Y_{TP}$). Thus, $Y_{TP}$ was the response of the design. This was achieved by optimizing the values of three key process variables, the DES/water proportion (%, *w/w*), the extraction time (*t*, min), and the extraction temperature (*T*, °C). Optimization was based on an experiment with a Box–Behnken design with 15 design points, including 3 central points. The ranges of the values of the process variables considered were chosen on the basis of previous investigations [12,13]. The process variables were set in 3 levels, −1, 0, and 1, according to the experimental design, and codified as reported in an earlier study [14]. The coded and actual levels are given in Table 1. Overall model significance ($R^2$, *p*), as well as and the significance of model (equations) coefficients, was assessed by ANOVA and lack-of-fit tests, at a minimum level of 95%.

**Table 1.** The independent variables used for the process optimization and their corresponding actual and coded levels.

| Independent Variables | Code Units | Coded Variable Level | | |
|---|---|---|---|---|
| | | −1 | 0 | 1 |
| DES/water (%, *w/w*) | $X_1$ | 65 | 75 | 85 |
| *t* (min) | $X_2$ | 20 | 120 | 220 |
| *T* (°C) | $X_3$ | 50 | 65 | 80 |

### 2.6. Total Polyphenol Determination and Antioxidant Tests

The protocols for the analysis of total polyphenols, antiradical activity ($A_{AR}$), and ferric-reducing power ($P_R$) have been given analytically in a previous publication [15]. In short, total polyphenols were determined with the Folin–Ciocalteu reagent and expressed as mg gallic acid equivalents (GAE) per g dry mass (DM), using a gallic acid calibration curve. The $A_{AR}$ was determined with the stable DPPH radical as the chromophore probe, and results were given as μmol DPPH per g DM. Determination of $P_R$ was carried out using TPTZ and expressed as μmol ascorbic acid equivalents (AAE) per g DM.

### 2.7. Chromatographic Analyses

Tentative identification of some compounds detected in the extracts generated was accomplished by liquid chromatography with diode array-mass spectrometry. Relevant detailed information regarding the equipment used and the conditions implemented has been provided in previous examinations [16,17]. For the quantification of compounds in the extracts, the device and protocols previously reported were employed [17,18].

### 2.8. Statistical Processing

All extraction processes were repeated at least twice, and all determinations were performed at least in triplicate. The values reported represent means ± standard deviation. The experimental setup concerning the design of the experiment and the response surface methodology, and distribution analyses, were elaborated by the JMP™ Pro 13 (SAS, Cary, NC, USA) software. Linear regressions were performed with SigmaPlot™ 12.5 (Systat Software Inc., San Jose, CA, USA). For all statistical analyses, a significance level of at least 95% (*p* < 0.05) was considered.

## 3. Results and Discussion

### 3.1. Rationale of DES Synthesis

The conceptual basis for designing a highly efficient DES was the utilization of food-grade, inexpensive substances, combined in a manner that would readily produce a liquid, which would be stable over time. On such a ground, the selection of DES constituents relied on previous observations, concerning the hydrogen bond donor(s) and the hydrogen

bond acceptor as well as their molar ratio. Glycerol was chosen as one of the HBDs because of its low price, high availability, and its wide range of applications in DESs, which proved it to be an efficient HBD [9]. Citric acid was included in the eutectic mixture, firstly because it provided a low pH, and secondly because it facilitated the formation of a eutectic mixture at increased HBA (amino acid) proportions. This was an important issue in the light of recent examinations, where evidence suggested that acidic DESs might be more efficient in polyphenol extraction [13]. Proline was preferred as an HBA because of the solid evidence that it might form DESs of increased effectiveness towards polyphenol extraction [19–21].

Considering all the above, the tailoring of a DES by combining glycerol, citric acid, and proline was attempted at various molar ratios. Initially, a molar ratio of glycerol:citric acid:proline 1:1:1 was tested but the constituents exhibited no satisfactory interaction as there was no formation of a perfectly transparent liquid. Switching the ratio to 2:1:1 was a successful combination, yielding a clear liquid, which when stored at ambient conditions over several weeks was stable, with no formation of crystals. This DES, termed as GL-CA-Pro, was used for further testing. To assess the performance of this DES, two other DESs were used: glycerol/choline chloride at a molar ratio of 2, termed as GL-ChCl, and glycerol/trisodium citrate at a molar ratio of 15, termed as GL-TSC. These two liquids were selected because they were both glycerol-based too and have been previously shown to perform well with regard to polyphenol extraction [13,22,23].

### 3.2. Screening of Various Plant by-Products

Initially, a screening was carried out including some common food by-products from Mediterranean plants, to examine their polyphenolic richness and to better assess the performance of the newly designed GL-CA-Pro. In Figure 1 it can be seen that the extraction of RGP, with any of the DESs used, afforded significantly higher $Y_{TP}$ values ($p < 0.05$), and the same held true for OLL extracted with GL-ChCl and GL-CA-Pro, but not GL-TSC. Results concerning $P_R$ exhibited exactly the same pattern (Figure 2B), while the $A_{AR}$ determination affirmed that extracts of both RGP and OLL produced with GL-CA-Pro were the most active (Figure 2A). Based on this outcome, it was evident that (i) both RGP and OLL were the wastes possessing the highest polyphenolic load and antioxidant activity, and (ii) GL-CA-Pro was the most effective solvent. Thus, RGP and OLL were further considered for process optimization, using GL-CA-Pro as the solvent of preference.

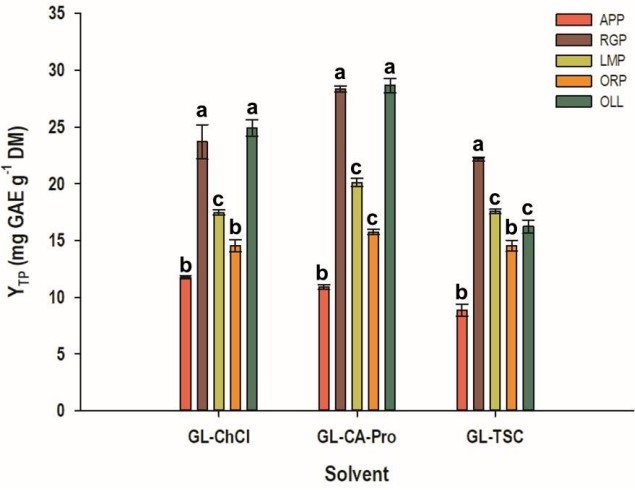

**Figure 1.** Performance of the three DESs tested (GL-ChCl, GL-CA-Pro, GL-TSC) on the extraction of total polyphenols from various agri-food wastes. The extractions were carried out at 50 °C for 180 min. Assignments are as follows: APP, apple peels; RGP, red grape pomace; LMP, lemon peels; ORP, orange peels; OLL, olive leaves. Columns denoted with different letters represent statistically different values ($p < 0.05$).

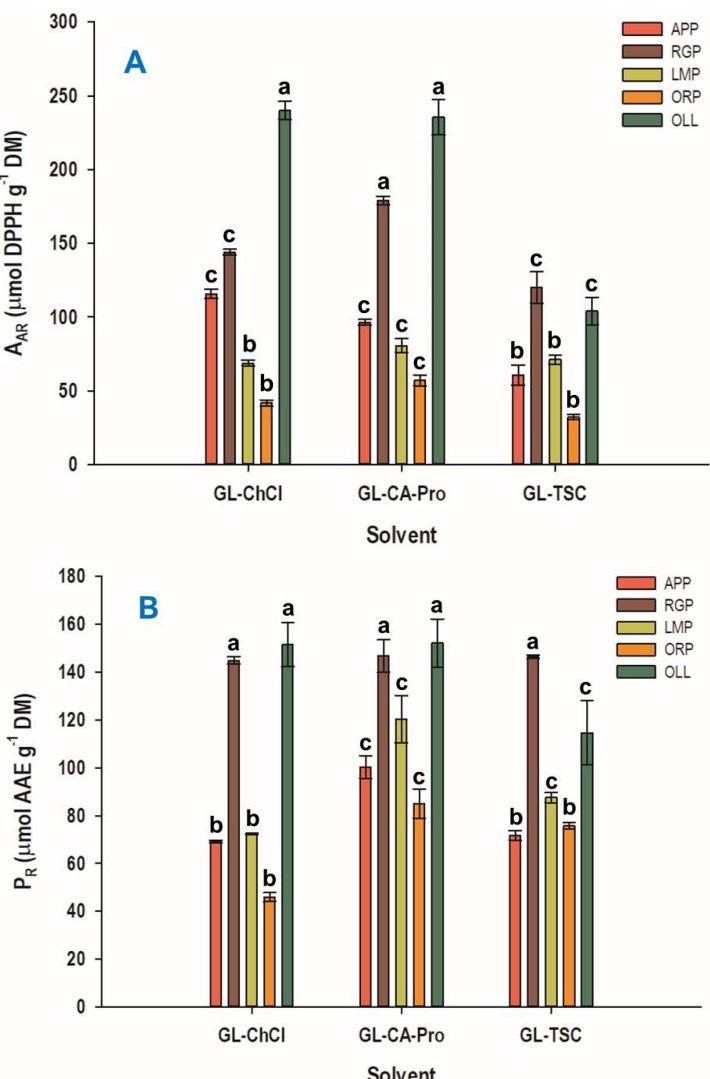

**Figure 2.** Antiradical activity (**A**) and ferric-reducing power (**B**) of the extracts produced with the three DESs tested (GL-ChCl, GL-CA-Pro, GL-TSC). The extractions were carried out at 50 °C for 180 min. Assignments are as follows: APP, apple peels; RGP, red grape pomace; LMP, lemon peels; ORP, orange peels; OLL, olive leaves. Columns denoted with different letters represent statistically different values ($p < 0.05$).

### 3.3. Process Optimization

The aim of the optimization process was to find a set of conditions that would meet the objective of the study, which was to attain a maximum response ($Y_{TP}$) in the preset interval of conditions. The response surface methodology was thus deployed to determine the optimum operating conditions. To this end, the effect of DES/water proportion (% *w/w*), termed as *C*, as well as the extraction time, *t*, and temperature, *T*, were considered and investigated. These are the most influential variables for an extraction process using a DES, as illustrated by previous studies [12,13,17,24]. The response surface suitability and model fitting assessment were based on ANOVA and lack-of-fit tests, considering the proximity of the measured and predicted values (Table 2). The coefficients determined for each model, the second-degree polynomial equations (models), and the statistical parameters are given in Table 3.

**Table 2.** Optimization of the extraction of RGP and OLL with GL-CA-Pro: the design points included in the experimental design, the corresponding coded values of the process variables, and the values for the measured and predicted responses.

| Design Point | Independent Variables | | | Response ($Y_{TP}$, mg GAE $g^{-1}$ DM) | | | |
|---|---|---|---|---|---|---|---|
| | $X_1$ (% *w/w*) | $X_2$ (*t*, min) | X3 (*T*, °C) | RGP | | OLL | |
| | | | | Measured | Predicted | Measured | Predicted |
| 1 | −1 | −1 | 0 | 23.86 | 23.86 | 26.39 | 27.91 |
| 2 | −1 | 1 | 0 | 26.20 | 28.23 | 32.82 | 32.06 |
| 3 | 1 | −1 | 0 | 17.02 | 14.97 | 21.84 | 22.61 |
| 4 | 1 | 1 | 0 | 34.66 | 34.66 | 40.68 | 39.16 |
| 5 | 0 | −1 | −1 | 21.00 | 22.77 | 10.05 | 7.93 |
| 6 | 0 | −1 | 1 | 23.31 | 23.58 | 28.74 | 28.53 |
| 7 | 0 | 1 | −1 | 21.78 | 21.51 | 20.87 | 21.09 |
| 8 | 0 | 1 | 1 | 50.67 | 48.90 | 34.03 | 36.07 |
| 9 | −1 | 0 | −1 | 22.33 | 20.57 | 19.87 | 20.42 |
| 10 | 1 | 0 | −1 | 18.29 | 18.56 | 23.76 | 25.06 |
| 11 | −1 | 0 | 1 | 34.16 | 33.89 | 43.26 | 41.96 |
| 12 | 1 | 0 | 1 | 31.67 | 33.43 | 39.66 | 39.11 |
| 13 | 0 | 0 | 0 | 20.13 | 21.36 | 36.75 | 38.12 |
| 14 | 0 | 0 | 0 | 21.16 | 21.36 | 40.62 | 38.12 |
| 15 | 0 | 0 | 0 | 22.78 | 21.36 | 37.00 | 38.12 |

**Table 3.** The mathematical models derived from the deployment of the response surface methodology, for the optimization of the extraction of RGP and OLL with GL-CA-Pro: only the significant terms were included in the models.

| Material | Second Order Polynomial Equations (Models) | $R^2$ | *p* |
|---|---|---|---|
| RGP | $21.36 + 6.02X_2 + 7.05X_3 + 3.83X_1X_2 + 6.65X_2X_3 + 3.33X_2{}^2 + 4.51X_3{}^2$ | 0.98 | 0.0014 |
| OLL | $38.12 + 5.18X_2 + 8.90X_3 + 3.10X_1X_2 − 7.96X_2{}^2 − 6.76X_3{}^2$ | 0.98 | 0.0011 |

Considering that $R^2$ provides an indication of the total variability around the mean estimated by the regression model, then for both RGP and OLL processes that $R^2 = 0.98$ it can be argued that the estimation of regression equations exhibited an excellent adjustment to the experimental data. In addition, assuming a confidence interval of 95%, lack-of-fit was not significant for either model, suggesting that the fitted models could be used for reliable predictions. Based on the models built, 3D graphs were constructed to depict the effect of process variables on the responses (Figures 3 and 4).

For RGP, the terms $X_2$ (*t*) and $X_3$ (*T*) were highly significant ($p < 0.001$), which highlighted their importance to the optimization of the process (Figure S1, inset "Parameter estimates" table). In contrast, the term $X_1$ (*C*) was not significant ($p > 0.05$), a fact that evidenced the low direct influence of DES/water proportion, within the limits tested, on $Y_{TP}$ maximization. Yet, the significance of the cross term $X_1X_2$ was high ($p < 0.05$) and positive, pointing to a synergistic effect between the composition of the solvent and the duration of the extraction process. It should be emphasized that the mass ratio of DES/water should always vary within certain limits. This is because a low amount of water (<20%) usually renders a DES inefficient for polyphenol extraction, most probably because the majority of DESs reported in the literature have high viscosity, which should be regulated by appropriate mixing with water. On the other hand, a water amount higher than 50% might cause disintegration of the DES and loss of its peculiar properties [25,26].

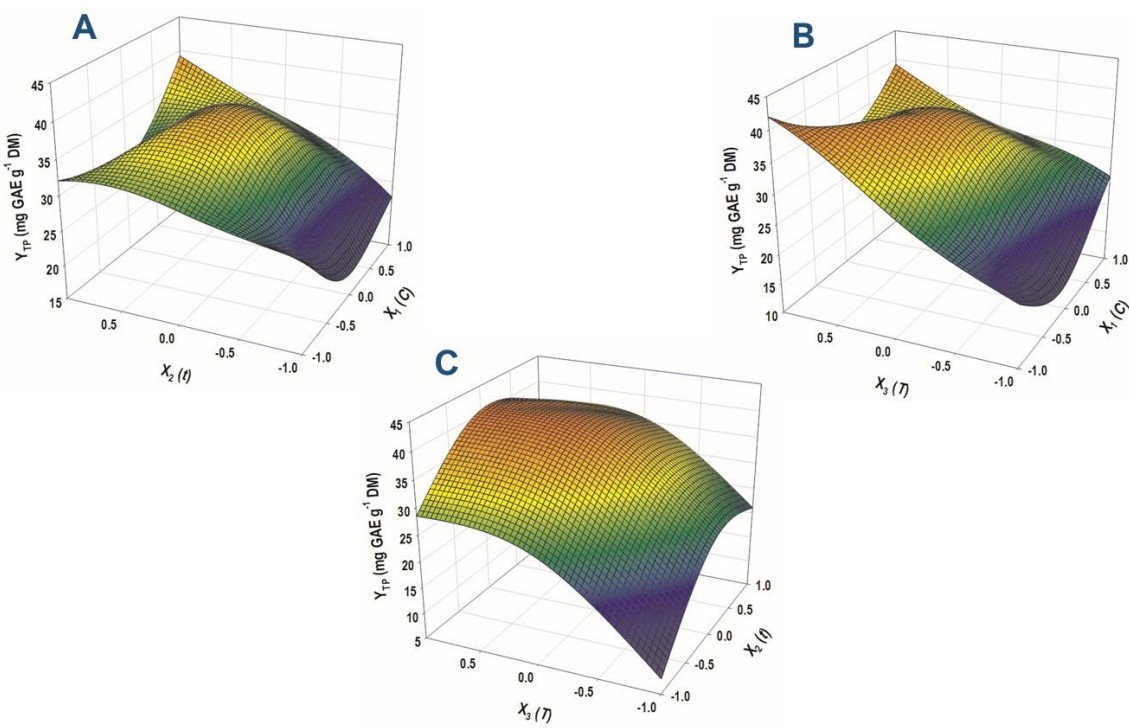

**Figure 3.** 3D graphs depicting the effect of the process variables considered on the response, for the extraction of RGP with GL-CA-Pro. Plot (**A**), covariation of $X_1$ (*C*) and $X_2$ (*t*); plot (**B**), covariation of $X_1$ (*C*) and $X_3$ (*T*); plot (**C**), covariation of $X_2$ (*t*) and $X_3$ (*T*).

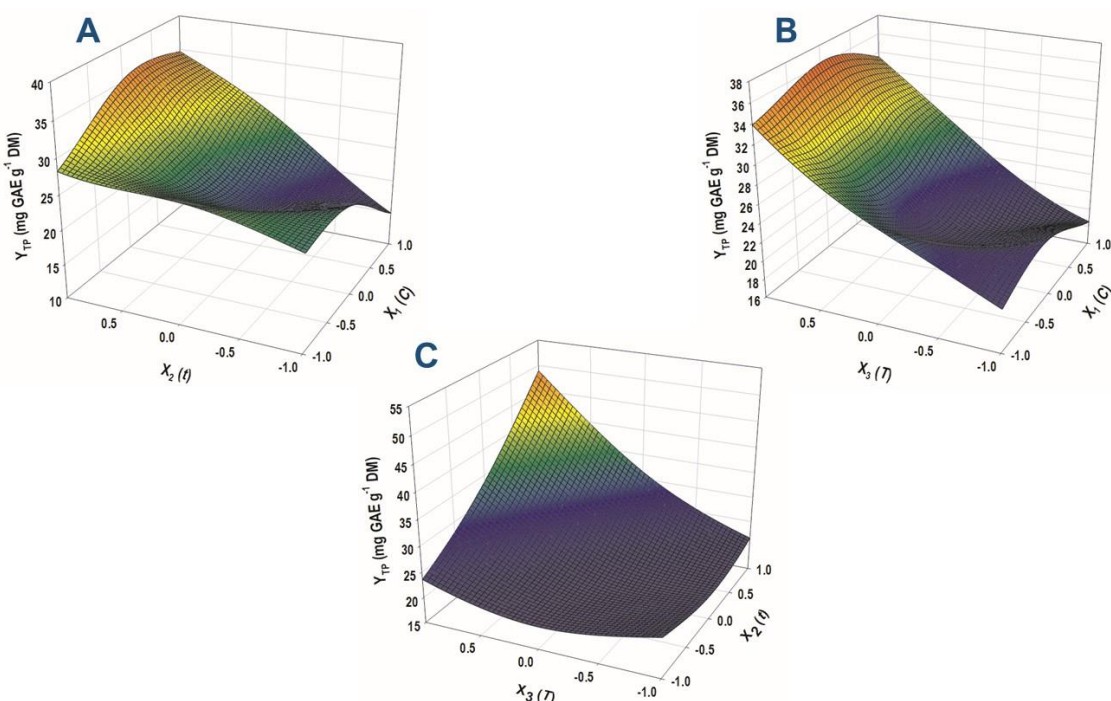

**Figure 4.** 3D graphs depicting the effect of the process variables considered on the response, for the extraction of OLL with GL-CA-Pro. Plot (**A**), covariation of $X_1$ (*C*) and $X_2$ (*t*); plot (**B**), covariation of $X_1$ (*C*) and $X_3$ (*T*); plot (**C**), covariation of $X_2$ (*t*) and $X_3$ (*T*).

The information revealed from the optimization of the extraction of OLL was very similar, in that $X_2$ (*t*), $X_3$ (*T*) and the cross term $X_1X_2$ were significant ($p < 0.05$) (Figure S2, "Parameter estimates" table). However, the negative effect of the quadratic terms $X_2$ (*t*) and

$X_3$ (*T*) indicated that increases in extraction period and temperature beyond an optimum value could have a negative impact on $Y_{TP}$ maximization.

The optimum levels for all three variables considered, as well as the maximum predicted values for $Y_{TP}$, were estimated by using the desirability function (Figures S1B and S2B). It can be seen in Table 4 that for both RGP and OLL, the optimum DES/water ratio was 85% (*w/w*) and the corresponding optimum *T* values were 80 and 72 °C. However, a large difference was found for the optimal extraction times, which were 220 and 168 min for RGP and OLL extraction, respectively.

**Table 4.** Maximum predicted responses and optimized conditions for the extractions of RGP and OLL with GL-CA-Pro. Predictions were based on the desirability functions (see Figures S1 and S2).

| Solvent | Maximum Predicted Response (mg GAE g$^{-1}$ DM) | Optimal Conditions | | |
| --- | --- | --- | --- | --- |
| | | % DES | *t* (min) | *T* (°C) |
| RGP | 53.25 ± 6.73 | 85 | 220 | 80 |
| OLL | 42.48 ± 4.19 | 85 | 168 | 72 |

The optimum *t* and *T* determined in this study for RGP were in very close agreement with recent data, reporting values of 240 min and 80 °C, respectively [18]. Yet, other processes involving RGP polyphenol extraction with DESs were shown to be less demanding, requiring 50 min and 65 °C [27]. With reference to OLL, the optimum level estimated for *T* (72 °C) was in line with previously published data, suggesting 73–75 °C as optimum [24,28]. Somewhat higher *T* (80 °C) has also been suggested by other studies using DESs as solvents [29,30]. According to a recent systematic study on optimal *t* and *T* for polyphenol extraction, the average corresponding values for an extraction with a DES were 210 min and 68 °C [12].

Comparison of the efficiency of the DES with that of two green conventional solvents, water and 60% aqueous ethanol, showed that RGP extraction with DES may afford 380% higher $Y_{TP}$ compared to water, which was found to be the second most efficient solvent (Figure 5). For the extraction of OLL, the DES was also the most efficient solvent, providing 69% higher $Y_{TP}$ compared to aqueous ethanol.

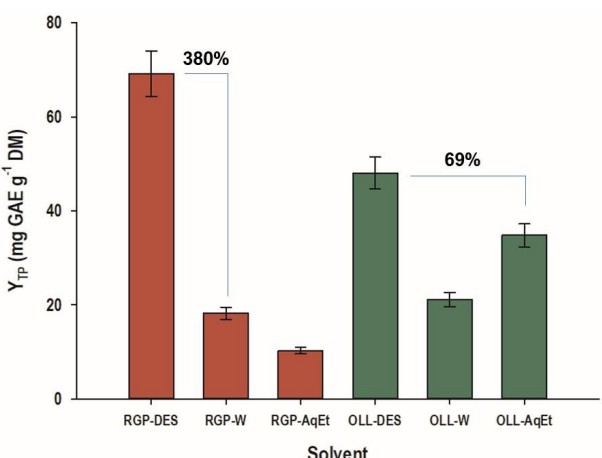

**Figure 5.** Comparison of the performance of GL-CA-Pro (assigned as DES) with those of 60% aqueous ethanol (AqEt) and water (W), with regard to polyphenol extraction from RGP and OLL. The extractions with DES were performed under optimized conditions, as given in Table 4. The extractions with water and aqueous ethanol were carried out at 70 °C, for 60 and 180 min, respectively.

### 3.4. Polyphenolic Profiles

The extracts of both RGP and OLL produced under optimized conditions were analyzed with HPLC to profile the principal polyphenolic substances. With reference to

RGP, the chromatogram at 280 nm revealed the presence of gallic acid and catechin, while monitoring at 320 and 360 nm also showed the presence of caftaric acid (caffeoyltartaric acid), rutin (quercetin 3-*O*-rutinoside), and quercetin (Figure 6). The identification of all compounds was based on the comparison of retention times and UV-vis spectra with those of original standards, with the exception of caftaric acid. This compound was identified on the basis of LC-MS analysis, as reported elsewhere [31].

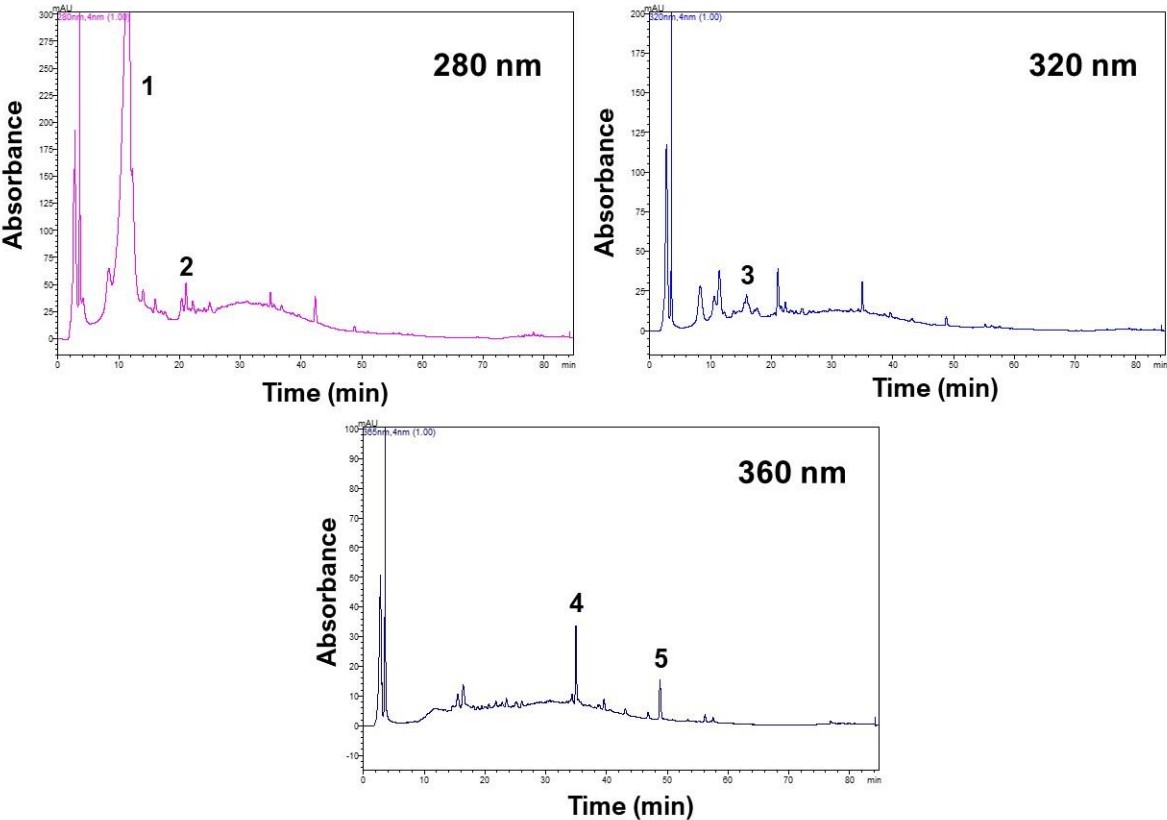

**Figure 6.** Chromatograms of an RGP extract produced with GL-CA-Pro, under optimized conditions given in Table 4. Peak assignments are as follows: 1, gallic acid; 2, catechin; 3, caftaric acid; 4, rutin; 5, quercetin.

The quantitative analysis shown in Table 5 demonstrated that the extraction with DES (GL-CA-Pro) exhibited selectivity particularly for catechin, whose yield was 14.8-fold higher than that achieved with water and aqueous ethanol, respectively. A slightly higher yield (1.89-fold) was also seen for gallic acid in the extract obtained with a DES compared to water. To the contrary, the hydroethanolic extract had the highest rutin yield. Overall, the yield of the extraction afforded with a DES was 1457.54 µg g$^{-1}$ DM, being 3.33- and 2.65-fold higher than that attained with water and aqueous ethanol, respectively.

Regarding OLL, eight principal constituents were considered (Figure 7), and peak identification was based on previously reported data [17]. In this case, the supremacy of using the DES as the solvent was shown only for hydroxytyrosol, whereas for the recovery of all the other compounds aqueous ethanol was the most effective solvent. More particularly, extraction with DES gave a hydroxytyrosol yield of 243.74 µg g$^{-1}$ DM, which was 2.25-fold higher compared to that of aqueous ethanol extraction. On the other hand, aqueous ethanol was significantly more effective for oleuropein extraction, giving a 1.80-fold higher yield compared to DES extraction (Table 6).

**Table 5.** Quantitative analysis of major polyphenols detected in RGP extracts. Assignments and extraction conditions are those reported in Figure 5. Values given are mean ± standard deviation (sd).

| Compound | Yield (µg g$^{-1}$ DM) | | |
|---|---|---|---|
| | **Water** | **AqEt** | **DES** |
| Gallic acid | 208.43 ± 0.46 | 182.81 ± 7.02 | 394.54 ± 1.06 |
| Caftaric acid | 29.47 ± 0.23 | 30.07 ± 0.70 | 33.03 ± 0.26 |
| Catechin | 41.64 ± 0.04 | 52.32 ± 3.20 | 772.57 ± 22.91 |
| Rutin | 147.11 ± 0.19 | 239.43 ± 0.35 | 210.39 ± 0.52 |
| Quercetin | 11.44 ± 0.03 | 44.40 ± 1.00 | 47.02 ± 0.27 |
| *Sum* | *438.10* | *549.03* | *1457.54* |

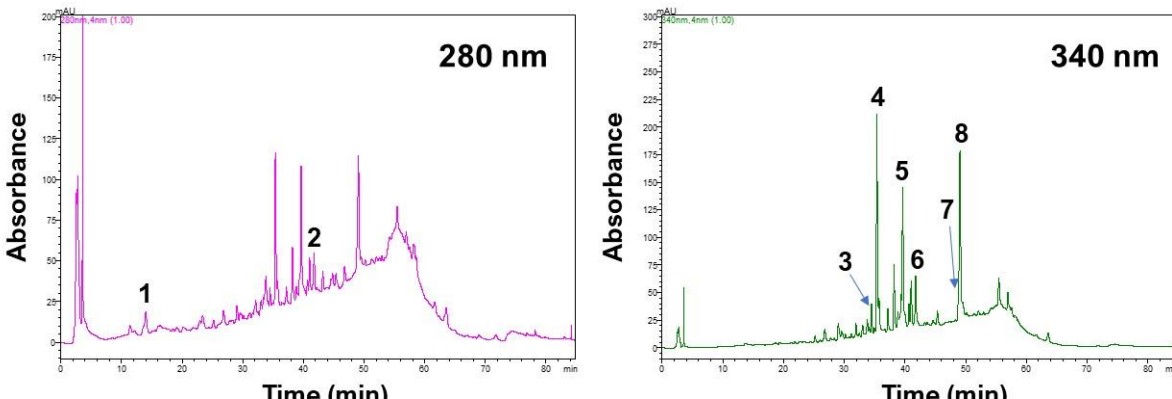

**Figure 7.** Chromatograms of an OLL extract produced with GL-CA-Pro, under optimized conditions given in Table 4. Peak assignments are as follows: 1, hydroxytyrosol; 2, oleoropein; 3, rutin; 4, luteolin 7-*O*-glucoside; 5, apigenin 7-*O*-rutinoside; 6, luteolin 3-*O*-glucoside; 7, quercetin; 8, luteolin.

**Table 6.** Quantitative analysis of major polyphenols detected in OLL extracts. Assignments and extraction conditions are those reported in Figure 5. Values given are mean ± standard deviation (sd).

| Compound | Yield (µg g$^{-1}$ DM) | | |
|---|---|---|---|
| | **Water** | **AqEt** | **DES** |
| Hydroxytyrosol | 55.86 ± 3.63 | 108.34 ± 0.45 | 243.74 ± 7.63 |
| Rutin | 69.46 ± 0.19 | 170.68 ± 0.83 | 160.63 ± 1.12 |
| Luteolin 7-*O*-glucoside | 139.04 ± 1.46 | 997.19 ± 1.40 | 897.36 ± 0.65 |
| Apigenin 7-*O*-rutinoside | 27.16 ± 0.41 | 256.51 ± 0.08 | 225.56 ± 0.35 |
| Luteolin 3′-*O*-glucoside | 42.98 ± 0.6 | 217.60 ± 3.31 | 198.00 ± 1.71 |
| Oleuropein | 573.93 ± 4.00 | 1721.02 ± 28.45 | 958.17 ± 20.68 |
| Quercetin | nd | 279.91 ± 4.35 | 96.15 ± 4.07 |
| Luteolin | 144.79 ± 5.42 | 1040.14 ± 8.88 | 872.55 ± 4.34 |
| Apigenin | 5.21 ± 0.02 | 83.64 ± 0.50 | 70.57 ± 0.56 |
| *Sum* | *977.62* | *3834.89* | *2834.14* |

### 3.5. Antioxidant Properties

The RGP extract produced with the DES was 95% higher $A_{AR}$ compared to the aqueous extract, which was the second most active in this regard (Figure 8A). Likewise, the corresponding $P_R$ values had an almost equal difference of 93% (Figure 8B). The pattern observed for the OLL extracts was different, as the DES extract had lower $A_{AR}$ compared to the hydroethanolic extract by 5.5%.

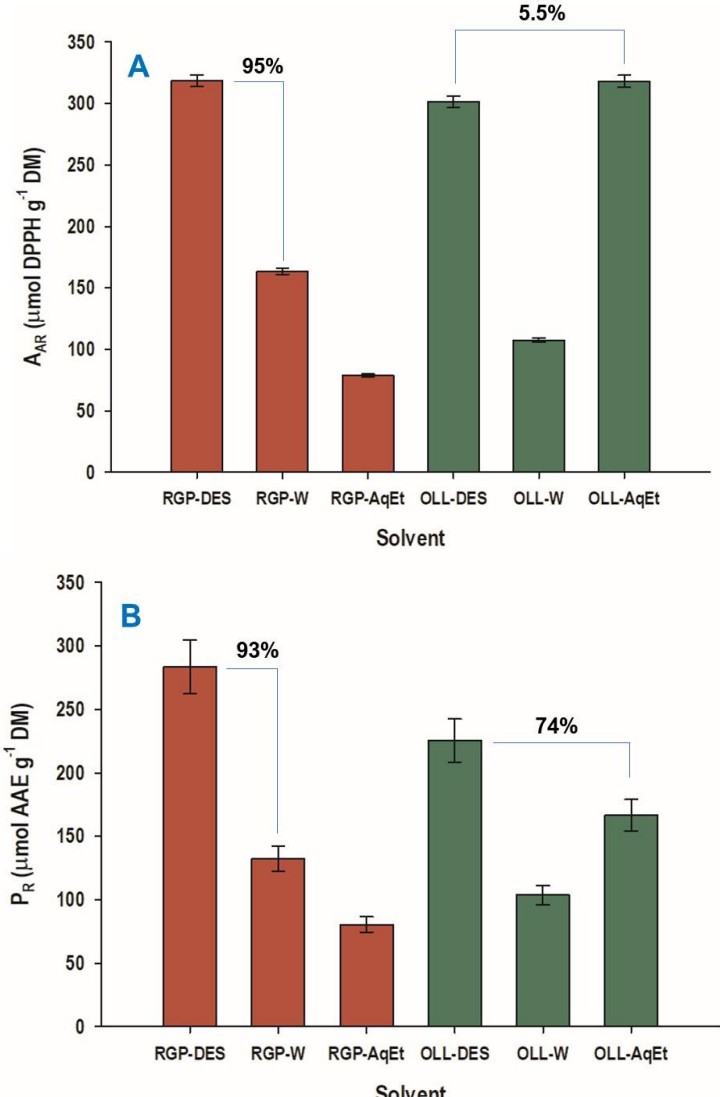

**Figure 8.** Comparison of the antiradical activity (plot **A**) and ferric-reducing power (plot **B**) of the RGP and OLL extracts obtained with GL-CA-Pro (assigned as DES), with those of 60% aqueous ethanol (AqEt) and water (W). The extractions with DES were performed under optimized conditions, as given in Table 4. The extractions with water and aqueous ethanol were carried out at 70 °C, for 60 and 180 min, respectively.

However, this difference was statistically negligible. On the other hand, the DES extract displayed a $P_R$ value that was 74% higher than the corresponding one for the hydroethanolic extract. These differences were likely reflected the polyphenolic composition of the extracts. It would appear that the high levels of gallic acid and catechin in the DES extract of RGP (Table 5) significantly enhanced both its $A_{AR}$ and $P_R$, hence its stronger antioxidant properties. Gallic acid and catechin have been demonstrated to exert very powerful antiradical activity compared to several other common phenolic acids, such as caffeic or ferulic acid [32]; therefore, it could largely define the antioxidant behavior of the extracts.

In the case of OLL, it has been proposed that the antioxidant activity is mainly expressed by major substances, including oleuropein, oleuropein derivatives, and luteolin 7-*O*-glucoside [33,34]. Thus, it might be argued that the differences found for those constituents in the OLL extract profiles reflected the antioxidant characteristics. It should be emphasized that, overall, the antioxidant activity was the integration of the antioxidant effects exerted by the individual polyphenols, as well as the manifestation of antagonistic/synergistic effects amongst them [35,36]. On this ground, it would be rather

impossible to make predictions for actual final effects based merely on the concentration of specific compounds.

## 4. Conclusions

The main objectives of this study were first, to rationally synthesize a novel, tailor-made, tertiary DES; second, to test its efficiency with regard to polyphenol extraction on several agri-food waste materials; and third, to optimize the extraction process for the most polyphenol-rich wastes. Thus, this was attempted in the design of a novel, food-grade tertiary DES, composed of glycerol, citric acid, and L-proline (GL-CA-Pro). The initial screening of several abundant agri-food wastes indicated that this particular DES was more effective than other widely used DESs, and that RGP and OLL were the two materials that gave extracts enriched in polyphenols, with increased antioxidant activity. Optimization of the extraction process demonstrated that the optimal extraction time might be significantly different for those two materials, whereas the optimal DES/water ratio was identical. The chromatographic analyses also revealed that there might be important differences in principal polyphenolic constituents among extracts produced with DES and water or aqueous ethanol. Such differences were claimed to reflect on the antioxidant properties of the extracts. As a final conclusion, it could be said that the novel DES tested was shown to be a high-performance system for the extraction of polyphenols from various agri-food wastes, and specifically for RGP and OLL polyphenol extraction. On the other hand, regarding OLL, its efficiency was lower for most of the main substances compared to aqueous ethanol.

**Supplementary Materials:** The following supporting information can be downloaded at: https://www.mdpi.com/article/10.3390/su14116864/s1, Figure S1: Desirability function (graph A), and plot of predicted vs actual values of the response ($Y_{TP}$) (plot B), for the optimization of the extraction of RGP polyphenols performed with GL-CA-Pro. Inset tables provide statistics associated with the assessment of the model derived. Values with color and asterisk are statistically significant; Figure S2: Desirability function (graph A), and plot of predicted vs actual values of the response ($Y_{TP}$) (plot B), for the optimization of the extraction of OLL polyphenols performed with GL-CA-Pro. Inset tables provide statistics associated with the assessment of the model derived. Values with color and asterisk are statistically significant.

**Author Contributions:** Conceptualization, S.I.L. and D.P.M.; data curation, V.A. and D.P.; formal analysis, V.A., D.P., K.P. and E.B.; methodology, V.A. and D.P.M.; project administration, S.I.L. and D.P.M.; supervision, S.I.L. and D.P.M.; writing—original draft, V.A., S.I.L. and D.P.M.; writing—review and editing, S.I.L. and D.P.M. All authors have read and agreed to the published version of the manuscript.

**Funding:** This research was co-financed by the European Union and the Hellenic Ministry of Economy & Development through the Operational Program Competitiveness, Entrepreneurship, and Innovation, under the call RESEARCH—CREATE—INNOVATE (project code: T2EDK-03772).

**Institutional Review Board Statement:** Not applicable.

**Informed Consent Statement:** Not applicable.

**Data Availability Statement:** The data used to support the findings of this study are available from the corresponding author upon request.

**Acknowledgments:** The authors acknowledge and appreciate the contribution of the Green Sustainable Innovation Research Group (Department of Food Science & Nutrition) in performing some of the experiments of this study.

**Conflicts of Interest:** The authors declare no conflict of interest.

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
