# Peer review of "Extraction of Polyphenolic Antioxidants from Red Grape Pomace and Olive Leaves: Process Optimization Using a Tailor-Made Tertiary Deep Eutectic Solvent"

_sustainability, doi:10.3390/su14116864_

Round 1

Reviewer 1 Report

Manuscript presented for review with title: “Extraction of polyphenolic antioxidants from red grape pomace and olive leaves: process optimization using a tailor-made tertiary deep eutectic solvent” is interesting and very important.

The experiment was planned very carefully. I’m impress of excellent work made by authors. The Introduction section includes all necessary information about examined objects and problems.

The collected experimental material and used methods do not raise any objections.

The discussion section presents a very good comparison of the obtained results with other results available in the data basis.

General opinion: I think that presented manuscript is a very valuable with extremely high scientific value and should be published in presented form in Sustainability journals.

Author Response

We sincerely acknowledge the time and effort the reviewer dedicated to review our paper.

Reviewer 2 Report

It is a comprehensive study that attempted to design a novel, food-grade tertiary DES, composed of glycerol, citric acid and L-proline (GL-CA-Pro), and optimize an extraction process for the effective recovery of polyphenolic compounds from residual food biomass (red grape pomace and olive leaves). The study appears to be performed with care. The experiments were adequately performed and presented, the results were interpreted appropriately, and the discussion was with clarity. 

A minor recommendation- spelling errors, article usage, and misspelled words should be checked and corrected.

Author Response

We sincerely acknowledge the time and effort the reviewer dedicated to review our paper.

Regarding spelling errors mentioned by the reviewer, we took every effort to improve our manuscript by extensively revising the text, to detect and correct possible errors.

Reviewer 3 Report

In the abstract section: The background should be focused on introducing the main objective of the study before the objectives. In the methods, please just mention the main layout. Why red grapes with olives leaves? What is the link between them? More information about the highly significant observations should be clarified. You should provide Conclusion with focusing on the innovations related to be in a harmony with the title of the manuscript and the objectives of your study.

Generally, introduction section needs several enhancements and error checks. I think the language should be rechecked by an English expert for grammar and typos. In the introduction, more information about the used techniques for the extraction like deep eutectic solvents (DES), and the other used methods for extraction and identification like TLC and GC/FID and MS benefits for this study should be clarified. You should focus on the objectives of your study. Introduce why your methods are important with recent references.

Why you specially used grape to compare it with olive leaves?

Please clarify why you used ferric-reducing power compared to DPPH method? How could you ensure that the condition was optimized and the compounds were stabled?

In the results: Why you did not apply the same methods with chemical solvent for comparing the condition and yields? Clarify this point.

In conclusion: More details about your observations by showing some of this valuable study significations point by point should be more clarified.

Author Response

We sincerely acknowledge the time and effort the reviewer dedicated to review our paper.

  1. In the abstract, a general background of the objectives was added, as instructed by the reviewer.
  2. The reason for examining in depth red grape pomace and olives leaves is adequately explained in section 3.2. In this section it is clearly mentioned that RGP and OLL were shown to be the richest (with regard to polyphenols) agri-food wastes, for which the novel DES synthesized performed very well in terms of polyphenol extract. This is the reason why those two materials were chosen for further investigation. Furthermore, those two materials were not compared to each other, but they were both used to bring out the extraction efficiency of the novel DES.
  3. With regard to the comment of the reviewer about the control extractions with conventional solvents, it should be clarified that the comparison was not based on the conditions (time, temperature) using different solvents, but on the different methodologies (different set of condition). Thus, the optimized methodology of RGP and OLL extraction was compared to methodologies using water and aqueous ethanol, under time and temperature conditions proposed as optimum in a recent study (Morsli et al., 2021).
  4. The two antioxidant tests used (FRAP, DPPH) were performed as complementary assays and not to compare them with each other. These two tests are abundantly used by many researchers and provide an estimation of the antioxidant activity of the extracts produced. In any case, the results from those two tests can be used as an additional criterion to compare the methodology developed with the control extractions, and so we did.
  5. In the Introduction section, we added a paragraph to analytically illustrate the main objectives of the study, as proposed by the reviewer.
  6. The Conclusions section was modified to better point out the main objectives of the study and the main findings, as suggested by the reviewer.
  7. With regard to further improving the Introduction section, we took every effort to detect and correct any syntax and spelling errors. On the other hand, we oppose the proposal of the reviewer concerning the report on methodologies about TLC and GC/MS identification of the extract compounds. This is because i) there are no reports in the literature concerning RGP or OLL extract analysis using TLC and ii) the same holds true for GC/MS, since the latter technique is used for volatile identification. In our case no volatiles were analysed; to the contrary, all compounds detected were non-volatiles.

Round 2

Reviewer 3 Report

-